# An Easy-To-Use External Fixator for All Hostile Environments, from Space to War Medicine: Is It Meant for Everyone’s Hands?

**DOI:** 10.3390/jcm12144764

**Published:** 2023-07-19

**Authors:** Julie Manon, Vladimir Pletser, Michael Saint-Guillain, Jean Vanderdonckt, Cyril Wain, Jean Jacobs, Audrey Comein, Sirga Drouet, Julien Meert, Ignacio Jose Sanchez Casla, Olivier Cartiaux, Olivier Cornu

**Affiliations:** 1Université Catholique de Louvain (UCLouvain), 1348 Louvain-la-Neuve, Belgium; michael.saint@uclouvain.be (M.S.-G.); jean.vanderdonckt@uclouvain.be (J.V.); jean.jacobs@uclouvain.be (J.J.); audrey.comein@uclouvain.be (A.C.); sirga.drouet@student.uclouvain.be (S.D.); julien.meert@student.uclouvain.be (J.M.); jose.sanchez@student.uclouvain.be (I.J.S.C.); olivier.cornu@saintluc.uclouvain.be (O.C.); 2Morphology Lab (MORF), UCLouvain—IREC, 1200 Brussels, Belgium; 3Neuromusculoskeletal Lab (NMSK), UCLouvain—IREC, 1200 Brussels, Belgium; 4Orthopedic Surgery Department, Cliniques Universitaires Saint-Luc, 1200 Brussels, Belgium; 5Crew 227—Mission Analog Research Simulation (M.A.R.S. UCLouvain), Mars Desert Research Station (MDRS), Hanksville, UT 84734, USA; cyril.wain@hotmail.com; 6European Space Agency, Blue Abyss, Newquay TR8 4RZ, UK; pletservladimir@gmail.com; 7Department of Health Engineering, ECAM Brussels Engineering School, Haute Ecole “ICHEC-ECAM-ISFSC”, 1200 Brussels, Belgium; crt@ecam.be

**Keywords:** tibial shaft fracture, external fixator, hostile environments, learning curve, space, developing countries, war medicine

## Abstract

Long bone fractures in hostile environments pose unique challenges due to limited resources, restricted access to healthcare facilities, and absence of surgical expertise. While external fixation has shown promise, the availability of trained surgeons is limited, and the procedure may frighten unexperienced personnel. Therefore, an easy-to-use external fixator (EZExFix) that can be performed by nonsurgeon individuals could provide timely and life-saving treatment in hostile environments; however, its efficacy and accuracy remain to be demonstrated. This study tested the learning curve and surgical performance of nonsurgeon analog astronauts (*n* = 6) in managing tibial shaft fractures by the EZExFix during a simulated Mars inhabited mission, at the Mars Desert Research Station (Hanksville, UT, USA). The reduction was achievable in the different 3D axis, although rotational reductions were more challenging. Astronauts reached similar bone-to-bone contact compared to the surgical control, indicating potential for successful fracture healing. The learning curve was not significant within the limited timeframe of the study (N = 4 surgeries lasting <1 h), but the performance was similar to surgical control. The results of this study could have important implications for fracture treatment in challenging or hostile conditions on Earth, such as war or natural disaster zones, developing countries, or settings with limited resources.

## 1. Introduction

Long bone fractures are common musculoskeletal injuries that, while they can be easily managed in developed countries, can become a whole different story when they occur in hostile or uncommon conditions. Fractures may have increased risks of complications such as bleeding, infection, and delayed healing due to the unique conditions present in those environments, such as space, war or natural disaster zones, developing countries, or settings with limited resources. The problem of having a long bone fracture in hostile or uncommon and challenging environments is that traditional methods of fracture repair may not be feasible or effective due to various challenges. These challenges may include weightlessness, restricted resources, absence of healthcare facilities, difficulties in soft tissue and wound management, challenges in anesthesia administration, and, above all, limited access to surgical expertise [1]. Therefore, finding appropriate and effective methods for fracture repair in hostile environments is crucial to ensure successful healing, functional recovery, patient outcomes, and survival, while not compromising all other activities that depend on that injured person.

As a first step, the external fixator can already solve some of these challenges. External fixation allows preservation of fracture hematoma and management of soft tissues, is less invasive, reduces bleeding and infection risk, and could require only a local or locoregional anesthesia [2,3,4]. A correctly executed procedure could potentially allow immediate weight-bearing, which is crucial for mission success, soldier autonomy, and faster consolidation compared to casts.

However, a major limitation in hostile environments is the restricted availability of trained surgeons to set up the fixator despite the huge need [1]. For example, more than 21% of the surgical activities of Médecins Sans Frontières (MSF) are orthopedic surgeries, including external fixators [1,2,3,4,5], but are not usually performed by a specialized surgeon. The fixator can decrease the rate of amputation and enhance limb salvage and life in humanitarian contexts [5]. Nonsurgeon individuals may need to be self-sufficient and autonomous in managing fractures in such situations, like during space missions, on the battlefield of war or natural disaster, or in remote areas with limited access to medical care. Fractures may also occur as part of emergency situations where immediate intervention is needed to stabilize the fracture and prevent further life-threatening complications. Nonsurgeon individuals who would be trained in using fracture fixation methods could provide timely and life-saving treatment in such situations, even in the absence of skilled surgical personnel. Therefore, having an easy-to-use external fixator (EZExFix) that can be performed by nonsurgeon individuals could allow for prompt, effective, and self-sufficient treatment of fractures without relying solely on surgical expertise.

In this study, our newly developed EZExFix [6,7] is designed to be easy, quick to learn, and accessible to nonsurgeon individuals as a solution for stabilizing tibial shaft fractures, which are one of the most common types of long bone fractures [8,9,10].

Space was chosen as the ultimate hostile environment to test the efficacy and accuracy of the management of tibial shaft fractures by nonsurgical astronauts. With space exploration missions extending beyond Earth orbit, such as potential travel to Mars, the health and safety of astronauts become critical [2,11,12,13,14,15,16]. The absence of an orthopedic surgeon in space, combined with the occurrence of a long bone fracture, poses serious risks to the health and life of the injured astronaut and may jeopardize the entire mission. Repatriation for timely surgical treatment is unfeasible due to the vast distance between Mars and Earth. Telesurgery has limitations due to significant transmission delay [2,13,17,18,19]. Therefore, it is crucial to enhance the autonomy of astronauts, empowering them with the skills and resources necessary to effectively manage medical emergencies on their own and to achieve enough medical outcomes.

Analog astronauts had to quickly learn how to assemble the EZExFix in order to fix tibial shaft fractures during a simulated inhabited mission at the Mars Desert Research Station (MDRS) lasting 2 weeks. This station is a simulated Martian habitat located in the Utah desert, USA [20]. Every year since 2002, from November through April, it serves as a research facility for studying human factors and conducting experiments relevant to future Mars missions [21]. The station provides a Mars-like environment and allows scientists and astronauts to simulate living and working conditions on the Red Planet [22]. The effectiveness and accuracy of fracture reduction, i.e., the medical performance of nonsurgeon astronauts, are evaluated independently without relying on Earth support or extensive surgical skills. This assessment includes analyzing the learning curve based on four surgical sessions (Sessions S_1_, S_2_, S_3_, S_4_) and whether some extreme conditions such as simulated extravehicular activities (S_EVA_) or at an unexpected moment (S_stress_) can alter their performance. The final objective is to utilize these findings to draw conclusions that can be extrapolated to Earth for the treatment of fractures in challenging or hostile conditions.

## 2. Materials and Methods

### 2.1. Fractured Leg Model

The fractured leg model was already described in a previous article [4]. Briefly, the leg model was made from a left tibia with a hard cortical and cancellous intramedullary bone structure (LSH1385, Synbone SDN BHD, Kulai, Malaysia). The AO/OTA 42A2 fracture type was always created by marking a simple oblique fracture line in the middle of the tibial diaphysis using a laser and three vertical benchmarks for references (Figure 1). The leg model was shaped with foam rubber sheet (22320, Komprex^®^, Lohmann & Rauscher, Neuwied, Germany), covered with a sock to mimic the skin, and fixed onto a foot prosthesis to provide an axis for realignment maneuvers. A new leg model was used for each new surgery. The removal of soft tissues from the bone was possible without having to remove the EZExFix and allowed measurement of the quality of reduction.

### 2.2. External Fixator (EZExFix)

The EZExFix is a newly developed fixator designed to stabilize tibial fractures with an emphasis on the ease of use, the price, and the accessibility in hostile and challenging environments [6]. This device can fix all types of tibial shaft fractures, including complex or comminuted fractures with significant soft tissue lesions. It has been validated to have mechanical properties similar to the Hoffmann^®^ 3 fixator, which is a reference device [7]. The EZExFix consists of various spare parts that can be assembled into a final construct, which is illustrated in Figure 2.

### 2.3. Study Design

#### 2.3.1. Analog Surgeons

Six analog astronauts participating in the Tharsis mission (2022) at the MDRS were recruited for a study conducted in accordance with the hospital–faculty ethics committee of the Cliniques Universitaires Saint-Luc, Belgium (N°B403201523492). None of the analog astronauts was trained as surgeon, and none of them had experience treating long bone fractures before.

#### 2.3.2. Surgeries

At the early beginning of the mission, the analog astronauts received a brief theoretical training session lasting for one hour and a practical demonstration, during which they were taught about the indications, anatomical landmarks, and steps involved in using the EZExFix device.

The analog astronauts then competed with each other in a series of small timed runs where they had to set up the device on an artificial broken leg in the most efficient way, four times as operator who placed the EZExFix on the broken leg (Sessions S_1_, S_2_, S_3_, S_4_), and four times as assistant who helped to maintain the fracture reduction. The whole surgeries were performed without fluoroscopic control, the reduction being guided by the prosthetic foot and palpation of the anterior tibial crest. Each astronaut took turns being the operator or assistant in 12 rounds of runs, with each person being evaluated on four self-achievements (N = 24 experimentations) and on its learning curve. The timed runs were designed to simulate potential increasing stress levels in a challenging spatial environment where fractures may occur. To determine if stress levels could affect the performance, different learning conditions were used to induce stress. Each surgery was timed and carried out as a competitive trial between two operators. The surgeries were performed under three different conditions: standard, where all equipment was already prepared; stressful, which involved performing the surgery during an extravehicular activity (S_EVA_), or at an unexpected moment with no preparation (S_stress_). Each astronaut performed the surgeries twice under standard conditions and twice under stress conditions. This study design was already described in more detail previously [4].

#### 2.3.3. Surgical Control

Meanwhile, an experienced orthopedic surgeon also participated in the study as a surgical control, performing the same experiment as astronauts in standard conditions three times in order to compare astronauts’ surgeries to this reference point. The surgeon was experienced in using classical Hoffmann^®^ external fixators [23,24,25], but not the new EZExFix device, and received the same theoretical information as astronauts.

#### 2.3.4. Operating Schedule

Programming the surgeries for the EZExFix project among the eight different scientific projects of the analog mission was a complex combinatorial problem due to limited time and resources. To solve this, an artificial intelligence system called Romie was used to create a schedule for the entire mission and adapt it based on the progress of the mission in order to maximize the probability of mission success [26].

### 2.4. Analysis Parameters

#### 2.4.1. Data Collection for Fracture Reduction Positioning

Six main vectors are needed to characterize displacements between two fractured edges after reduction and fixation: three axes of translation following Cartesian coordinates (X, Y, and Z axes) and three of rotation (around each axis) (Figure 3a). In order to quantify them, a coordinate measuring machine (CMM) (Microscribe G2X, Immersion Corporation, San José, CA, USA) encoded the 3D position of a tip laying on points of interest with a precision of 0.2 mm (Figure 3b). First of all, baseline data of unbroken tibias were collected to model the “anatomical world” in order to allow the comparison with the “pathologic world” on fixed broken legs. To perform this, 20 points of tibial plateau were localized in the three Cartesian coordinates, including four cardinal points and 16 secondary points (Figure 3c; green dots). Then, three circumferences were added thanks to measuring three main points on the three tibial rims and 15 additional points in between on three different heights of the tibia (proximal, near the fracture, and distal) (Figure 3c; red dots). All those points allowed fitting of a cylinder approximating the correct tibial axis and which defined the Z axis. X and Y axes were then determined as orthogonal and according to main cardinal points. The X, Y, and Z axes define the world reference frame R_world_ with X the anteroposterior axis, Y the lateromedial axis, and Z the proximodistal axis. The fracture line was also measured in the anatomical position, before creating the fracture, described by three main points and 11 secondary points, and was used to separate the tibia into two parts (Figure 3c; blue dots).

After the break, the proximal part, above the fracture line, was considered as the reference in the pathologic world, i.e., fixed part that will be transposed in the anatomical world using the iterative closest point (ICP) registration algorithm using numerical computation software (MATLAB^®^, R2020b, The MathWorks, Natick, MA, USA). The distal part, below the fracture line, was seen as the moving part of the tibia which had to be reduced. Two local reference frames, R_p_ and R_d_, were defined on the proximal and distal parts of the fracture, respectively, using the 3D points encoded with the CMM (Figure 4). The result of the ICP registration between the proximal and distal parts of the fracture expressed in the pathologic world and the fracture line expressed in the anatomical world is the geometrical transformation that enables expression of the local frames R_p_ and R_d_ in the global frame R_world_ in terms of both position and orientation. After fixation, the same data were taken on each side of the fracture line and the residual shift between both could be calculated and compared with the anatomical world to translate into the six initial displacements, divided into three translational displacements and three rotational displacements. Translational displacements included sagittal translation or anteroposterior displacement (A/P), frontal translation or lateromedial displacement (L/M), and axial translation or shortening/lengthening (L−/L+). The rotational displacements included frontal rotation or varus/valgus (VR/VL), sagittal rotation or flessum/recurvatum (FL/RC), and axial rotation or external/internal rotation (ER/IR). Mathematically, those six parameters were calculated as follows. The A/P, L/M, and L−/L+ translational displacements were calculated as the distances in mm, along the X, Y, and Z axes of R_world_, respectively, between the positions of R_p_ and R_d_. The VR/VL, FL/RC, and ER/IR rotational displacements were calculated as the differences in orientation, in degrees, between R_p_ and R_d_ around the X, Y, and Z axes of R_world_, respectively.

#### 2.4.2. Quality Assessment of Fracture Reduction

All of the six displacements were considered pathologic above or below specific thresholds. The A/P, the L/M, and the L−/L+ displacements were pathologic outside the range of −5 to +5 mm of displacement. The VR/VL and FL/RC deformation were pathologic outside the range of −5° to +5° of rotation [27,28]. The ER/IR displacement were pathologic outside the range of −10° to +10° of rotation [27,28]. Outcomes are expressed by the mean of quantitative continuous variables and/or by a binary value if the astronaut reached the physiologic acceptable range or if he stayed in the pathologic one following the previous respective thresholds.

#### 2.4.3. Percentage of Bone Contact

In order to evaluate the surface of bone-to-bone contact, the minimal and maximal distances in mm between the proximal and distal parts of the fracture were calculated following the guidelines for an ISO1101-based assessment of the location parameter (Figure 5) [29]. Because it is widely accepted that the maximal gap to lead to direct bone healing through intramembranous ossification is 2 mm [30,31], this threshold was considered to describe the “bone-to-bone contact”. Larger gaps between bone ends may result in the formation of fibrous tissue instead of bone, leading to delayed healing or nonunion [32]. To refine the accuracy, a bone-to-bone contact under 1.5 mm was also assessed. The results are expressed as the percentage of bone-to-bone contact on all the tibial fracture circumference.

#### 2.4.4. Objective Learning Curve

The learning curve of nonmedical astronauts is essential to assess because of the significant impact on patient outcomes, surgical efficiency, and healthcare costs. This could also highlight the time needed for nonsurgeon personnel to handle the EZExFix. The four consecutive sessions allowed us to gain an idea of this short learning curve.

### 2.5. Statistical Analyses

A descriptive analysis was first performed to summarize and characterize data using boxplots. In the text, central tendencies are expressed by means, and dispersion by standard deviation (+/−SD) and by the ranges (min–max) where appropriate because of small datasets. Inferential statistics were used to make inferences and draw conclusions about population parameters. The normality of all data was evaluated by QQ plots. The comparisons between astronauts and the surgical control were assessed by nonparametric two-tailed Mann–Whitney tests. Nonparametric Friedman tests for repeated measures were computed to analyze differences among all sessions for each shift, followed by multiple Wilcoxon’s tests with Bonferroni correction if needed. The threshold for significance was set at 0.05, indicating that the null hypothesis would be rejected if the *p*-value was below. All statistical analyses were conducted using SPSS software (version 27, IBM SPSS, Inc., Chicago, IL, USA).

## 3. Results

### 3.1. Fracture Reduction—Translational displacement

Over the four sessions, translational mean shifts after analog astronauts’ surgeries were observed as follows: A posterior shift of −1.048 mm (+/−1.971; range −4.16–+2.25), a lateral shift of 3.247 mm (+/−3.119; range −2.33–+11.44), and a lengthening of −3.094 mm (+/−3.885; range −11.65–+5.16) (Figure 6). All the three translations had the same orientation as those of the surgeon, and none of these displacements were statistically different than that observed after the control surgeries: posterior shift of −1.320 mm (+/−3.750; range −4.74–+2.69) (*p* = 0.763), lateral shift of 0.420 mm (+/−2.909; range −2.83–+2.78) (*p* = 0.166), and a lengthening of −3.617 mm (+/−4.210; range −7.81–+0.61) (*p* = 0.966). None of the astronaut mean shifts was considered pathologic.

### 3.2. Fracture Reduction—Rotational Displacement

After the four surgeries, astronauts made an average varus displacement of 4.302° (+/−3.872; range −2.70–+10.05), a recurvatum of −12.284° (+/−11.479; range −34.80–+3.03), and an internal rotation of −9.860° (+/−8.078; range −24.51–+1.74) (Figure 7). The surgical control also ended with statistically indistinct varus of 3.340° (+/−3.873; range +0.82–+7.80) (*p* = 0.698) and recurvatum of −1.230° (+/−7.389; range −9.76–+3.20) (*p* = 0.094). However, the surgeon tended to place the foot in an external rotation of 2.920° (+/−4.179; range −0.27–+7.65), which was significantly different from the internal rotation created by astronauts (*p* = 0.008). While none of the residual displacements among the surgeon surgeries was clinically pathologic, the recurvatum was clearly pathologic for astronaut surgeries.

During the four sessions, each astronaut succeeded in reaching an average of almost four physiological axes (mean of 3.95 axes, +/−0.597). Translational reductions were easier to obtain than rotational ones. While the A/P physiologic range was systematically achieved (100% of astronauts), the L/M and the L−/L+ were obtained by 85 and 75% of astronauts, respectively. The VR/VL and the ER/IR success dropped down to 55 and 50%, respectively, but the biggest difficulty was obtaining a physiological range in the FL/RC (only 30% of astronauts reached the normal range on average over the four sessions).

### 3.3. Bone-to-Bone Contact

Concerning the cortical contact, the two bone ends were less than 2 mm from each other over 13.47% of the total tibial circumference for astronauts and over 20.60% for the surgeon (Figure 8). Despite this discrepancy, the difference was not significant (*p* = 0.60). However, when considering a bone-to-bone contact less than 1.5 mm apart, both surgeon and astronauts reached approximately 10% contact over the whole circumference (*p* = 0.166). Astronauts were able to successfully perform 60% of all the surgeries, with a minimal portion of 5% of the whole tibial circumference having contact less than 2 mm between the bones.

### 3.4. Objective Learning Curve

Figure 9 shows that across the four sessions, the A/P, the L−/L+, and the VR/VL tended to end with a better mean at S_4_ (closer to the zero baseline) but none of the six shifts showed a significant improvement (*p* > 0.05). The bony contact started to really improve at S_4_ but the difference with S_1_ was not yet significant (*p* = 0.109 both for contact < 2 and 1.5 mm). For half of the displacements (i.e., A/P, FL/RC, and ER/IR), the average data during EVA were better than that of S_stress_ and the opposite was seen for the remaining shifts (i.e., L/M, L−/L+, and VR/VL). The mean at S_4_ ended three times closer to the zero baselines than that of both S_EVA_ and S_stress_, suggesting that these conditions did not affect the performance of astronauts. The best clinical outcome at the end of S_4_ was the A/P reduction (0.51 mm +/−1.58; range −1.75–+2.25) and the worst one was the FL/RC reduction (−14.29° +/−13.74; range −34.80–+2.38).

## 4. Discussion

Having a reliable and effective fixation method for tibial fractures that can be handled by nonsurgeon astronauts is crucial to ensure proper bone healing and functional recovery in a hostile and challenging environment with limited access to medical facilities and resources.

The EZExFix succeeded in reaching these objectives and the reduction performance was relatively acceptable, although it needs to be discussed.

**Reduction displacements.** Translational reductions were easy to obtain, with an A/P reduction always obtained, probably due to the ease of palpation of the anterior tibial crest through very thin soft tissues, a true-to-life situation. However, the rotational ones were more complicated, with the FL/RC being the least successful one. Despite the apparent discrepancy in recurvatum between astronauts and surgeon, the difference was not significant (*p* = 0.094) because of the wide dispersion of data. This means that some surgeries were much better than others with non-negligible variability. The excessive variation resulted sometimes in pathologic recurvatum, which is clinically unacceptable. Although there is debate about the occurrence of osteoarthritis and functional outcomes after tibial malunion, a long follow-up study showed that even the malunion can increase the radiological osteoarthritis incidence, and it was not correlated with the patient joint symptomatology [28]. The only significant displacement was the axial rotation. The astronauts fixed the foot in IR while the surgeon fixed it more in ER (*p* = 0.008). Even if the IR remains in the physiologic range above −10° of rotation, the average value is quite borderline (−9.86° of IR). This can be viewed as clinically relevant also, because a residual ER is much more functional as residual IR [33,34]. Nevertheless, the general surgical residency (nonorthopedic surgery) does not give a predisposition to achieve direct correct alignment, because only two teams (2/6) succeeded in realigning a leg after a 2-day course [1].

**Bone-to-bone contact.** The improvement in bone-to-bone contact became noticeable at S_4_, but the difference compared to S_1_ was not statistically significant (*p* = 0.109). The bony contact is not significantly different between the surgeon and astronauts, which is a very good aspect of the healing prospect. The degree of contact is an important factor in fracture healing, as greater contact between the two bone ends facilitates bone growth and union. A higher percentage of contact generally indicates a better prognosis for healing, while a lower percentage of contact may increase the risk of delayed or nonunion [32].

**Objective learning curve.** Assessing the learning curve can aid in determining the number of cases required to become proficient and achieve consistent outcomes. This information can help to guide training programs and establish appropriate benchmarks for evaluating surgical proficiency [35], especially when the training time is restricted. It is also important for ensuring patient safety and optimizing the use of healthcare resources. Another study showed a rapid improvement for senior surgical residents who had never set up an external fixation before, but had some previous surgical knowledge [1]. Astronauts did not show any significant improvement in fracture reduction over the four sessions, suggesting that the number of surgeries may not have been sufficient to see a significant impact on the learning. However, despite the absence of a visible learning curve, the performance was similar to the surgical control for four displacements, meaning that perhaps there is no need for a long-duration learning curve. For the axial (ER/IR) and sagittal (FL/RC) rotation, there would be a real interest to repeat the study with more numerous sessions to see if astronauts could improve their learning curve on these rotational displacements that are more difficult to obtain, and to see how long it would take to meet physiological goals. Stressful conditions did not affect the astronauts’ performance, and a subjective learning curve was also assessed. The eight scales (i.e., attractiveness, efficiency, perspicuity, trust, adaptability, usefulness, intuitive use, and haptics) were consistently and continuously assessed over multiple sessions, indicating a pattern similar to a hype cycle curve.

**Patient safety, procedure, and time.** The fracture-related factors are needed to drive bone healing but patient safety, respect for the procedure, and the duration of execution are also three criteria important to maximize the treatment and were analyzed in a previous study [4]. As a brief reminder, in nearly all cases, safe zones, including arterial and nerve areas, were rigorously maintained to ensure the safety of patients during surgery. The steps of the procedure, the avoidance of skin compression, and the stability of the assembly were achieved in 80% of the surgeries. The average surgical time for an astronaut to apply an EZExFix (52.19 min +/− 11.08) was comparable to the surgical control and aligned with the mean operating time reported in the literature [36,37,38]. Significantly longer times were observed for reduction and fixation steps among astronauts, and the positioning step showed the most prominent difference comparing to the surgeon. Astronauts allocated more time to precise placement of the EZExFix and avoidance of skin compression, while surgeons prioritized the reduction step, possibly due to their experience. This indicates that astronauts quickly grasped the significance of these criteria, whereas surgeons may rely on their surgical expertise for the EZExFix positioning.

**Limits.** Tibial shaft fracture was used as a common long bone model and for its ease of handling because it allowed us to replicate exactly same fracture pattern and soft tissue condition. Consequently, the tibial crest is really easy to palpate and facilitates the self-monitoring of the reduction. This experience should be extended to other types or other bone fractures with more soft tissues around the bone and fewer direct benchmarks for reduction, such as femoral fractures. The small number of subjects due to the incompatibility of the space analog habitat with a large cohort is also a limit of this study. To address this, repeating the experiments in subsequent missions or in other hostile conditions on Earth could support the results obtained. The interpretation of all these results is also limited due to differences in variance between the surgical group, which was reduced to within-operator variance, and the astronaut group, which comprised both within- and between-operator variances. Therefore, caution should be exercised when comparing the astronauts’ performance with that of the surgeon, as it provides only an approximate idea of expected astronauts’ potential achievements. Additionally, future studies could include sterility evaluation. Although the EFORT (European Federation of National Associations of Orthopedics and Traumatology) open reviews have permitted the use of external fixators in emergency rooms for life-threatening patients with pelvic, humeral, femoral, or tibial instability [39], the EZExFix remains an aseptic surgical procedure that needs to be properly executed and assessed.

**Improvement and perspectives.** While the treatment of fractures in hostile and challenging environments is difficult, the diagnosis can be just as challenging too; however, the latter has to be firm before thinking about repairing a long bone fracture. Ultrasound has a well-established track record of accuracy in diagnosing abdominal and thoracic trauma, and it may be a valuable tool for diagnosing extremity injuries by trained nonphysician personnel in situations where radiography is not readily available, such as in military or space applications [40]. Once the diagnosis has been made, nonsurgeon individuals could be ready to use the EZExFix by having attended just one theoretical session lasting one hour, along with a single demonstration. However, in cases where there is any doubt about the reduction, the teacher should emphasize fixation of the leg in flexion and external rotation to counteract the average opposite tendency.

**Impacts on Earth medicine.** The parallelism between space and hostile environments on Earth is evident. War or natural disaster zones, developing countries, or settings with limited resources are facing the same challenges as previously described and need the same adaptations of the usual medicine or surgery. On the one side, medicine on Earth can help spatial medicine. For example, there is a wealth of collective maritime and naval experience, with a long-standing history of practices during times of war and peace, that could offer valuable insights on what actions to take and what to avoid when formulating space medicine policies [13]. But on the other side, the opposite is also valuable; space medicine or hurdles can help to improve Earth’s medicine in this kind of challenging situation. For example, some doctors from Médecins Sans Frontières (MSF) could be sometimes afraid about using external fixators to fix a fracture because it sounds like a complex procedure reserved for experienced surgeons, and they did not have sufficient exposure during their general surgery curriculum [1]. However, this study highlights the simplicity of the EZExFix and the possibility to achieve medical requirements without any medical background. This would encourage general practitioners, general trauma surgeons, or even nurses or personnel in the field to use the EZExFix in case of life-threatening emergencies or critical situations because of resource constraints. The medical evacuation procedures commonly used on Earth typically involve the application of a leg splint and transferring patients to healthcare facilities on the same day or within a short timeframe for appropriate medical care. This evacuation procedure could be optimized with this EZExFix in terms of fracture reduction and damage control surgery. However, immediate transportation is not always feasible, and relying solely on a splint in these cases does not address the need for definitive treatment, such as stable fixation of complex fractures with multiple fragments or dislocation, soft tissue management, or infection prevention in cases of open fractures. In situations where timely repatriation to a hospital is not possible, such as in developing countries, (sub)marines, or space missions, the lack of access to healthcare facilities can result in dire consequences. The implementation of the EZExFix provides a solution to these challenges, eliminating the dependency on healthcare facilities altogether. The EZExFix is made with materials that reduce the cost and allow local production in order to increase accessibility also for developing countries that are in need of easy accessible stabilization methods [41,42,43]. The EZExFix could be also considered for inclusion in a medical kit because it is compact, portable (requires only a battery for the drill; no external power supply needed), lightweight (1 kg without the drill), and takes up minimal space (35 × 20 × 10 cm, maximum of a shoebox) in the limited resources available such as on a spacecraft. This device could be readily available and could facilitate a rapid response in case of emergency, just like a defibrillator kit. The ultimate goal is to simplify the learning and utilization of the external fixator to alleviate apprehensions, reduce fears and preconceptions, and make it accessible and user-friendly, so that anyone can utilize it and save patients in challenging environments.

In conclusion, the EZExFix is a single device that offers the ability for nonsurgeon individuals to handle fracture fixation in hostile environments and to overcome challenges related to limited access to surgical expertise, emergency situations, resource constraints, self-sufficiency, and mission success or patient’s fate. Even if all the axes are not perfectly reduced, the consolidation should be achieved and the patient’s life saved by nonsurgeon people. The EZExFix combines nearly all the benefits needed to face challenging conditions, making it a promising orthopedic therapy for space applications as well as in settings with limited resources on Earth.

## Figures and Tables

**Figure 1 jcm-12-04764-f001:**
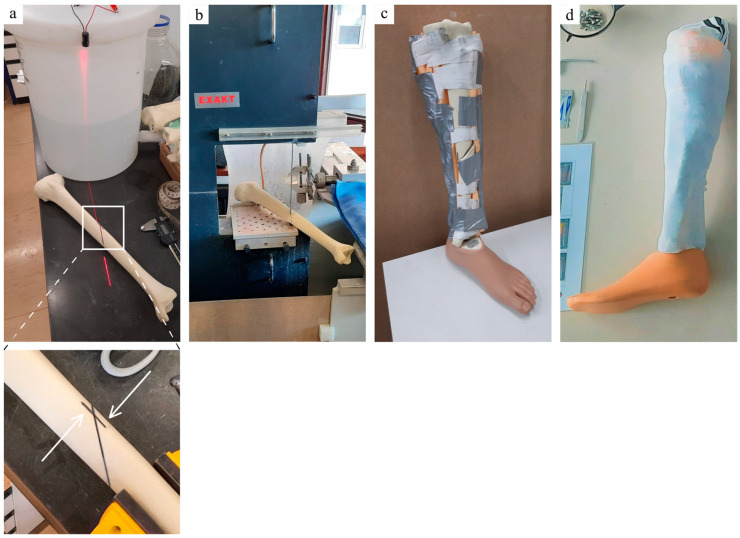
Creation of the fractured leg model. Landmark of the fracture line with a laser ((**a**) above—red line) and vertical benchmarks to further evaluate fracture reduction ((**a**) below—white arrows). Bone cutting by the diamond bandsaw following the fracture landmark (**b**). Soft tissues assembly and fixation around the fractured bone, mounted on a foot prosthesis (**c**). Final fractured leg model (**d**). Adapted from Manon et al. [4].

**Figure 2 jcm-12-04764-f002:**
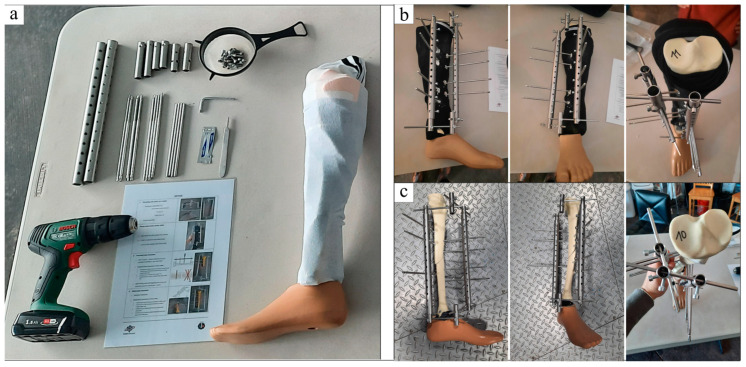
Material needed to build the EZExFix (**a**). Final construct mounted on a broken artificial leg on a sagittal, frontal, and upper view, respectively (**b**). Broken artificial leg after removing soft tissues ready to measure analysis parameters on a sagittal, frontal, and upper view, respectively (**c**). Adapted from Manon et al. [4].

**Figure 3 jcm-12-04764-f003:**
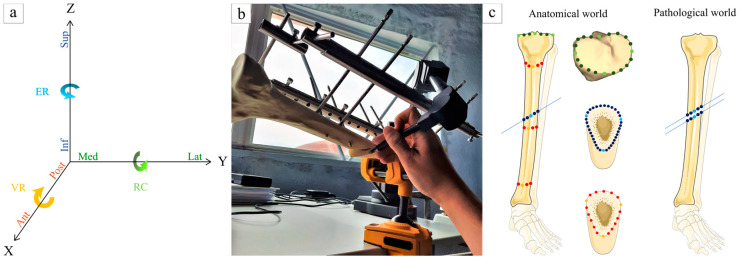
Measurement following six vectors (three translations and three rotations) into Cartesian coordinates (**a**). Ant: anterior, Post: posterior, Inf: inferior, Sup: superior, Med: medial, Lat: lateral, ER: external rotation, VR: varus, RC: recurvatum. For the procedure to harvest points position with the coordinate measuring machine, the pin has to point to the desired localization and a computer registers it (**b**). Points of interest to take measurements of anatomical world and pathologic world (**c**). The green and red dots are used to approximate the correct tibial axis and the blue ones are used to describe the fracture position.

**Figure 4 jcm-12-04764-f004:**
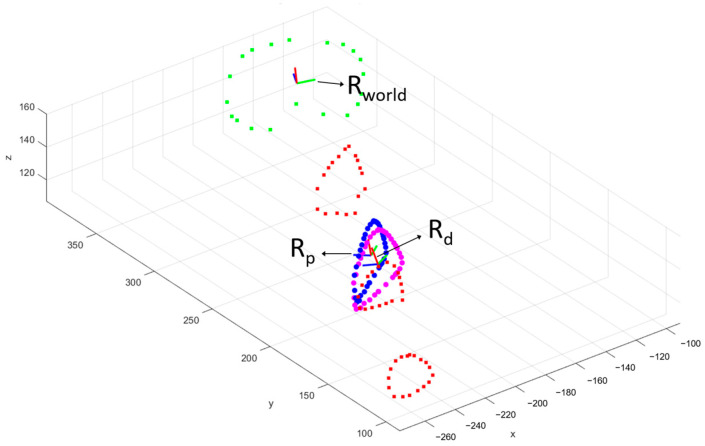
Examples of points of interest (in green, red, blue, and magenta) encoded with the coordinate measuring machine and used to construct the global reference frame R_world_ fixed to the tibial plateau (green dots), the local reference frame R_p_ fixed to the proximal part of the fracture (blue dots), and the local reference frame R_d_ fixed to the distal part of the fracture (magenta dots).

**Figure 5 jcm-12-04764-f005:**
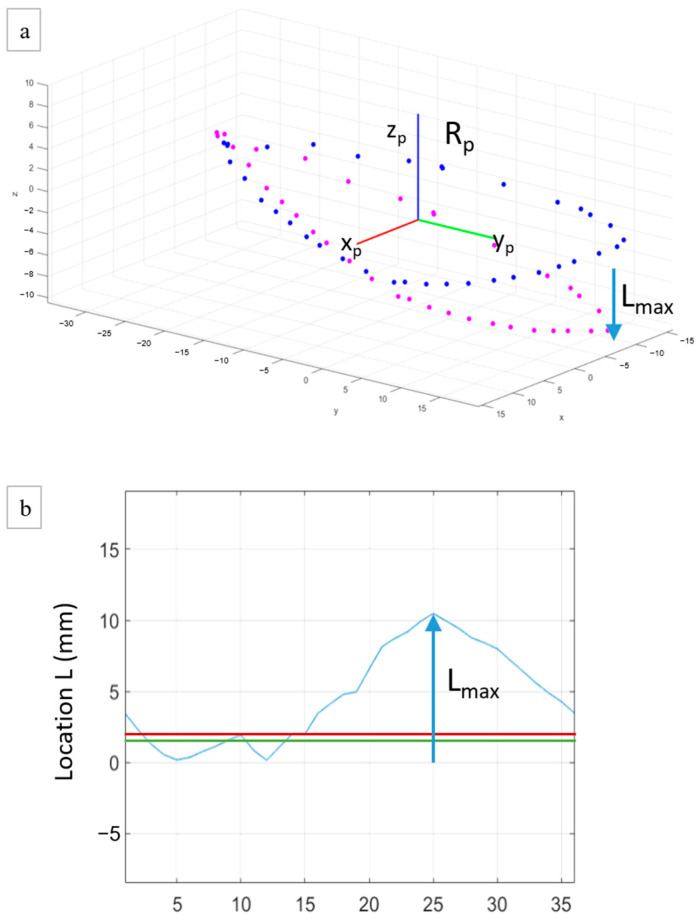
Example of proximal (blue dots) and distal (magenta dots) parts of the fracture registered in the global reference frame R_world_ (**a**). The local reference frame R_p_ is fixed to the centroid of the blue dots. As an illustration, L_max_ is the maximum distance between proximal and distal parts of the fracture, measured in mm along the axis z_p_. Example of results for the calculation of the location parameter between proximal and distal parts of the fracture (**b**). The blue curve represents the evolution of the location of the distal part along the circumference of the proximal part of the fracture. The horizontal red line is the 2 mm threshold. The horizontal green line the 1.5 mm threshold. The 2 mm and 1.5 mm bone-to-bone contacts are computed as the part of the blue curve lying under the red and green horizontal lines, respectively.

**Figure 6 jcm-12-04764-f006:**
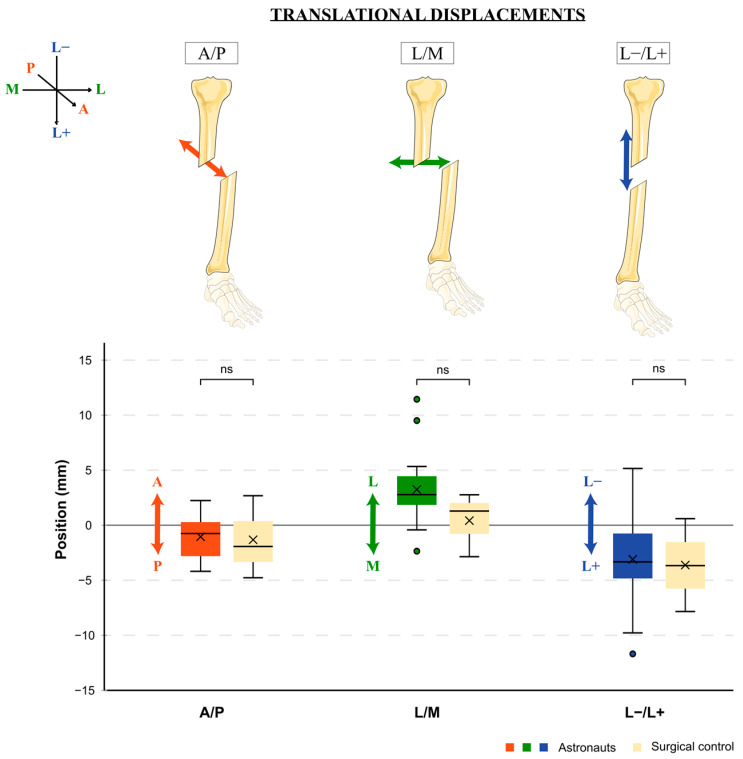
Average translational displacements of analog astronauts’ surgeries comparing to surgeon ones. A: anterior, P: posterior, M: medial, L: lateral, L+: lengthening, L−: shortening, ˟: mean, °: outliers, ns: nonsignificant.

**Figure 7 jcm-12-04764-f007:**
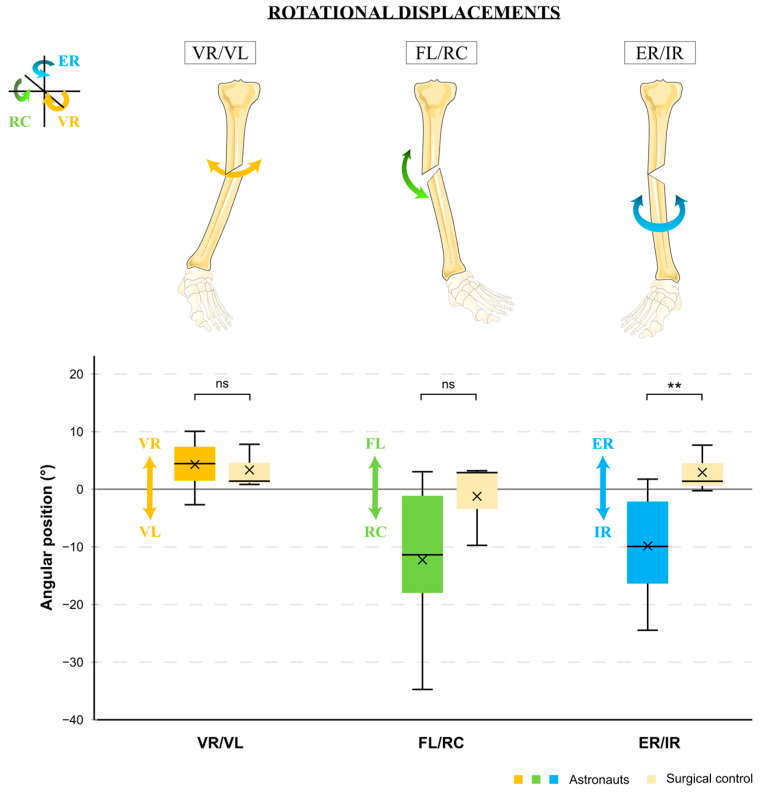
Average angular displacements of analog astronauts’ surgeries compared to surgeon ones. VR: varus, VL: valgus, FL: flessum, RC: recurvatum, ER: external rotation, IR: internal rotation, ˟: mean, ns: nonsignificant, **: *p* < 0.01.

**Figure 8 jcm-12-04764-f008:**
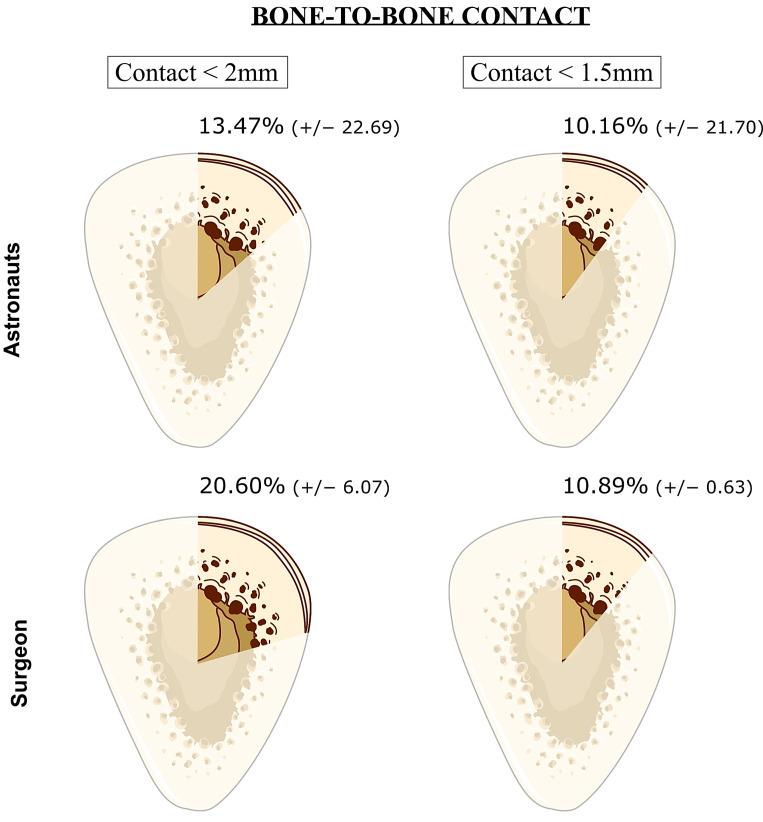
Bone-to-bone contact less than 2 and 1.5 mm for astronauts and the surgeon, represented on axial cross sections of the mid-shaft tibial fractures. The opaque surface corresponds to the bone contact percentage under the respective threshold. This contact zone is purely theoretical, not anatomical. Outcomes are expressed as the mean percentage over the total tibial circumference (+/−standard deviation).

**Figure 9 jcm-12-04764-f009:**
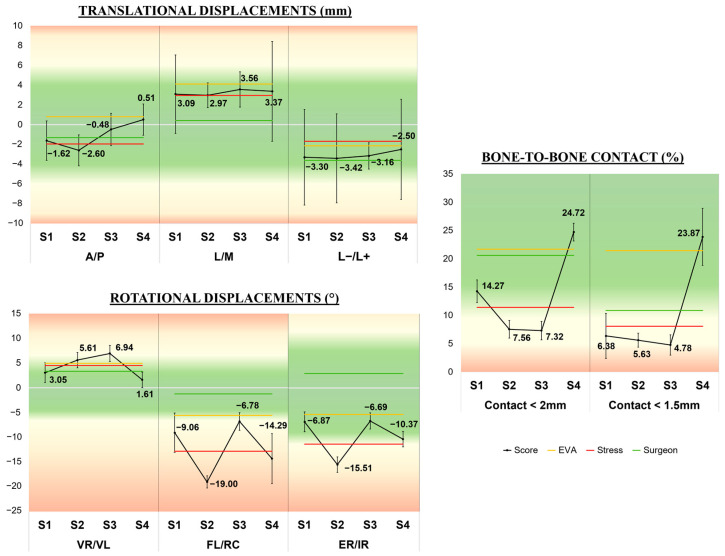
Panel chart of the six different displacements and the bone-to-bone contacts expressed as mean scores (black bold straight lines) across the four successive sessions (Sessions S_1_, S_2_, S_3_, S_4_). As a reference, S_EVA_ and S_stress_ are represented by yellow and red lines, respectively, as a horizontal bar calibrated on the corresponding mean scores. Error bars show the standard deviation. Green area shows physiologic range and red warns about the pathologic one. A: anterior, P: posterior, M: medial, L: lateral, L+: lengthening, L−: shortening, VR: varus, VL: valgus, FL: flessum, RC: recurvatum, ER: external rotation, IR: internal rotation.

## Data Availability

All data analyzed during the study are included in this published article. The complete original datasets generated during the current study are available from the corresponding author on reasonable request.

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
