# Peer review of "An Easy-To-Use External Fixator for All Hostile Environments, from Space to War Medicine: Is It Meant for Everyone’s Hands?"

_jcm, 2023, doi:10.3390/jcm12144764_

Round 1

Reviewer 1 Report

Thank you for submitting your recent study to your journal. The current manuscript deals with an interesting topic of applying easy-to-use external fixator (EZExFix) in an hostile environment. The manuscript is well structured and maintained with its focus on effectiveness of EZExFix on tibial shaft fracture by non-medical/non-surgeon personnel with proficient language of English. However, the reviewer would like to provide a few comments on the current study.

1. The study deals with applying EZExFix in an hostile environment, such as space and battlefields. However, the authors is curious what circumstances there would be when non-surgical personnel needs to perform ExFix on tibial shaft fracture because appliance of long leg splint, instead of ExFix, in such hostile environments without presence of surgeon seems to be a safer management plan for non-medical personnel than performing a surgical procedure. 

2. As mentioned in the limitation section, the non-medical personnel performing surgical procedure needs to take account of sterility, unavailability of radiologic assessment, and technical difficulties of the procedure. In the future study, the authors need to address other surgical factors that can affect the postoperative outcomes, rather than one's proficiency of surgical techniques. 

Author Response

Dear Reviewer,

On behalf of all the authors, I would like to thank you for the time and effort you dedicated to reviewing our manuscript titled “An easy-to-use external fixator for all hostile environments, from space to war medicine : Is it meant for everyone’s hands ?” as well as for your pertinent comments and suggestions. Please find below our point by point answers to your comments. Your initial feedback, copy-pasted in grey italic, is directly followed by the corresponding answer, written in normal black text.

Thank you for submitting your recent study to your journal. The current manuscript deals with an interesting topic of applying easy-to-use external fixator (EZExFix) in an hostile environment. The manuscript is well structured and maintained with its focus on effectiveness of EZExFix on tibial shaft fracture by non-medical/non-surgeon personnel with proficient language of English. However, the reviewer would like to provide a few comments on the current study.

  1. The study deals with applying EZExFix in an hostile environment, such as space and battlefields. However, the authors is curious what circumstances there would be when non-surgical personnel needs to perform ExFix on tibial shaft fracture because appliance of long leg splint, instead of ExFix, in such hostile environments without presence of surgeon seems to be a safer management plan for non-medical personnel than performing a surgical procedure.

Usually, splinting is a viable temporary treatment option during evacuation procedures with the aim of transferring patients to healthcare facilities on the same day or within a day for more appropriate care. However, immediate transportation is not always feasible, and this approach does not address the need for definitive treatment, such as stable fixation of complex fractures with multiple fragments or dislocation, soft tissue management, or infection prevention in cases of open fractures. In situations where timely repatriation to a hospital is not possible, such as in developing countries, (sub)marines, or space missions, the lack of access to healthcare facilities can result in dire consequences. The utilization of the EZExFix provides a solution to these challenges, eliminating the dependency on healthcare facilities altogether. This has been added in the discussion part.

  1. As mentioned in the limitation section, the non-medical personnel performing surgical procedure needs to take account of sterility, unavailability of radiologic assessment, and technical difficulties of the procedure. In the future study, the authors need to address other surgical factors that can affect the postoperative outcomes, rather than one's proficiency of surgical techniques.

Thank you for your advice. We fully agree with these perspectives, and we will ensure to address them accordingly.

We are very grateful to you for your comments and queries, which were quite thoughtful for us. Therefore, we have adapted the manuscript accordingly, so as to incorporate your suggestions. We modified the main text and manuscript by adding the previously explained modifications marked up using the “Track Changes” function.

Once again, thank you so much for allowing us to clarify the open issues.

Sincerely Yours,

The authors.

Reviewer 2 Report

Dear author, 

thank you for your really interesting paper. Here are my comments and suggestions:

Discussion should also include the possibility of MEDEVAC with secondary surgical treatments. One big point of this study is the applicability of surgical techniques that are often only used temporarily.

Sterility in austere environments, need for soft tissue surgery and radiographic techniques for nonmedical providers are already addressed and would also lead to evacuation procedures.

The shown fracture reduction would also suggest feasible reduction in austere environments, leading to a possibility of evacuation optimized for further care as well as damage control surgery (DCS), pain management and regaining some mobility for the patient in austere environments.

It's somewhat unclear from the paper if astronauts receive a basic medical and anatomical training in general before attending the study.

Include the weight of the equipment with the measurements in line 466. It's next to measurement, a great factor for portability in an austere environment setting, as well relevant to space medicine in terms of payload.

Language: Improve semantic consistency in naming and choosing the most professional term e.g. non-sugical providers/personnel/individuals/people.

Author Response

Dear Reviewer,

On behalf of all the authors, I would like to thank you for the time and effort you dedicated to reviewing our manuscript titled “An easy-to-use external fixator for all hostile environments, from space to war medicine : Is it meant for everyone’s hands ?” as well as for your pertinent comments and suggestions. Please find below our point by point answers to your comments. Your initial feedback, copy-pasted in grey italic, is directly followed by the corresponding answer, written in normal black text.

Dear author,

thank you for your really interesting paper. Here are my comments and suggestions:

  1. Discussion should also include the possibility of MEDEVAC with secondary surgical treatments. One big point of this study is the applicability of surgical techniques that are often only used temporarily. Sterility in austere environments, need for soft tissue surgery and radiographic techniques for nonmedical providers are already addressed and would also lead to evacuation procedures.

Indeed, this is also a good point to discuss and a little paragraph has been added in the discussion part.

  1. The shown fracture reduction would also suggest feasible reduction in austere environments, leading to a possibility of evacuation optimized for further care as well as damage control surgery (DCS), pain management and regaining some mobility for the patient in austere environments.

We completely agree with your statement.

  1. It's somewhat unclear from the paper if astronauts receive a basic medical and anatomical training in general before attending the study.

Page 4 mentioned : “None of the analog astronauts was trained as surgeon, and none of them had expe-rience treating long bone fractures before.

2.3.2.    Surgeries

At the early beginning of the mission, the analog astronauts received a brief theo-retical training session lasting for one hour and a practical demonstration, during which they were taught about the indications, anatomical landmarks, and steps in-volved in using the EZExFix device… … This study design was already described in more details previously [4].”

  1. Include the weight of the equipment with the measurements in line 466. It's next to measurement, a great factor for portability in an austere environment setting, as well relevant to space medicine in terms of payload.

It has also been added in the discussion.

  1. Comments on the Quality of English Language. Language: Improve semantic consistency in naming and choosing the most professional term e.g. non-sugical providers/personnel/individuals/people.

The different semantic names are used in order to differentiate surgeons that are legally certified, non-surgical personnel/providers that could be healthcare professional but without any surgical graduation, and non-surgical individuals/people that could be anyone, without any medical background at all. For example, astronauts are mostly engineers at the beginning without any medical background (before their space training).

We are very grateful to you for your comments and queries, which were quite thoughtful for us. Therefore, we have adapted the manuscript accordingly, so as to incorporate your suggestions. We modified the main text and manuscript by adding the previously explained modifications marked up using the “Track Changes” function.

Once again, thank you so much for allowing us to clarify the open issues.

Sincerely Yours,

The authors.